# BEAST: Efficient Tokenization of B-Splines Encoded Action Sequences for Imitation Learning

**Hongyi Zhou**[†][*]    **Weiran Liao**[†]    **Xi Huang**[†]    **Yucheng Tang**[†]    **Fabian Otto**[§]
**Xiaogang Jia** [†]    **Xinkai Jiang** [†]    **Simon Hilber** [†]    **Ge Li** [†]    **Qian Wang** [†]
**Ömer Erdinç Yağmurlu** [†]    **Nils Blank** [†,‡]    **Moritz Reuss** [†]    **Rudolf Lioutikov**[†,‡]
[†] Karlsruhe Institute of Technology    [‡] Robotics Institute Germany    [§] Microsoft Research

## Abstract

We present the **B**-spline **E**ncoded **A**ction **S**equence **T**okenizer (**BEAST**), a novel action tokenizer that encodes action sequences into compact discrete or continuous tokens using B-spline. In contrast to existing action tokenizers based on vector quantization or byte pair encoding, BEAST requires no separate tokenizer training and consistently produces tokens of uniform length, enabling fast action sequence generation via parallel decoding. Leveraging our B-spline formulation, BEAST inherently ensures generating smooth trajectories without discontinuities between adjacent segments. We extensively evaluate BEAST by integrating it with three distinct model architectures: a Variational Autoencoder (VAE) with continuous tokens, a decoder-only Transformer with discrete tokens, and Florence-2, a Vision-Language Model with an encoder-decoder architecture, demonstrating BEAST's compatibility and scalability with large pretrained models. We evaluate BEAST across three established benchmarks consisting of **166 simulated tasks** and on three distinct robot settings with a total of **8 real-world tasks**. Experimental results demonstrate that BEAST (i) significantly reduces both training and inference computational costs, and (ii) consistently generates smooth, high-frequency control signals suitable for continuous control tasks while (iii) reliably achieves competitive task success rates compared to state-of-the-art methods. Videos and code are available at https://intuitive-robots.github.io/beast_website/.

## 1   Introduction

Imitation learning has emerged as a powerful paradigm for training robots to perform complex tasks by learning from human demonstrations [1–3]. Early works [4, 5] in this field primarily focused on predicting single-step actions based on the current observation. However, recent research [6] highlights the importance of learning action sequences to capture the temporal coherence inherent in human demonstrations. Moreover, by modeling action sequences, we can reduce compounding errors [7] and create task demonstrations that more closely align with human methods [8]. Given the success of autoregressive next-token prediction models in natural language processing and other domains [9–11], it is compelling to explore similar techniques for modeling action sequences, leveraging their ability to predict and generate coherent sequences effectively.

In natural language processing, tokens typically represent words, which are inherently discrete elements. This discrete nature allows for effective next-token prediction, which extends well to the generation and prediction of symbolic actions or in discrete action space. However, a significant challenge arises when attempting to apply these approaches to sub-symbolic, continuous actions, which are not inherently discrete. Discretization addresses this issue by compressing the continuous

---

[*]Correspondence to `hongyi.zhou@kit.edu`

action sequence while trying to retaining essential information. This process helps in balancing the expressivity of the action representation against computational efficiency.

Despite growing interest in this area, effective strategies to create action sequences of discrete tokens remain underexplored. Existing approaches often focus on single-step tokenization [12–14], vector quantization [15–17], or compression-based schemes [18]. However, they require training separate encoder-decoder networks for the tokenizer [15, 19] or produce variable-length token sequences for inputs of the same duration [18], which complicates applying fast token generation techniques such as parallel decoding [20]. Furthermore, existing action tokenizers do not consider the smooth transitions between subsequent action chunks, which could result in undesired jumps at transition.

To address these challenges, we propose the B-spline Encoded Action Sequence Tokenizer (**BEAST**), a novel tokenizer that represents continuous action sequences using B-splines [21]. BEAST offers versatility, allowing for effective integration with both discrete and continuous tokens. Different from tokenizers based on the vector quantization [15–17], it does not require additional tokenizer training. BEAST compresses action trajectories into fixed-length token sequences enabling efficient parallel decoding for faster token generation, requiring $4 - 8\times$ fewer tokens than binning-based tokenization. By using B-spline encoded control points as discrete tokens, BEAST ensures the generation of smooth action chunks, as well as the continuous connection between consecutive chunks.

**Our contributions are:** 1) We introduce BEAST, a novel B-spline-based tokenizer designed for modeling continuous action sequences. 2) We demonstrate the versatility of BEAST by integrating it into diverse model architectures that accommodate both continuous and discrete objectives. 3) We conduct extensive evaluations of simulated and real-world robotic tasks, showcasing its effectiveness. 4) We perform thorough ablation studies to assess the impact of various design choices.

## 2    Related Work

Prior work has explored various action representations for policy learning. The most common approach is to directly predict low-level actions, such as joint positions or end-effector displacements, using a supervised learning objective [5, 4, 22]. While simple, these approaches cannot tackle the multimodality present in human behavior.

To address these limitations, ACT [6] introduces an Action Chunking Transformer trained as a conditional Variational Autoencoder (CVAE), which models multimodal behavior via a learned latent space. Instead of predicting single actions, ACT generates entire action chunks in a single inference step. These chunks are short sequences of actions, which reduces covariate shift and improves performance. Another line of work focuses on generating action sequences with diffusion models. Diffusion Policies model complex, multimodal behaviors by iteratively denoising from Gaussian noise to generate action sequences [7, 23–26]. While effective, these methods require multiple denoising steps per sequence, making inference comparatively expensive. In contrast, BEAST compresses full action sequences into compact control-point representations using B-spline approximation. This significantly reduces the number of predictions needed to model temporally extended behaviors. As a result, it enables efficient action chunking with smooth transitions, combining the representational benefits of ACT and diffusion policies with the speed and simplicity of tokenized inference.

A prominent research direction in action sequence representation is Movement Primitives (MPs) [27–30]. Dynamic Movement Primitives (DMPs) [27, 29] model trajectories using a second-order dynamical system with a goal attractor, ensuring smooth transitions in position and velocity. However, DMPs are limited in representing higher-order continuity. To address these limitations, recent works [30, 31] have explored using B-splines as an alternative representation for MPs. Despite their effectiveness, existing works have primarily applied MPs within reinforcement learning (RL) frameworks [32–36, 31] and only utilize them as continuous action representations.

Alternatively, robot actions can be represented as discrete values by discretizing them into a set of tokens. This discretization scheme is common in many recent Vision-Language-Action models (VLAs) [12, 37–40, 16, 17]. These models, often based on Transformers, are well-suited to predicting discrete tokens due to their autoregressive pretraining on language. A common discretization technique involves dividing the continuous action space into a fixed number of bins [41, 13]. However, this strategy struggles to effectively model high-frequency robot data. Further it has very low inference speed. More sophisticated tokenization methods have been proposed. Behavior Transformers [19]

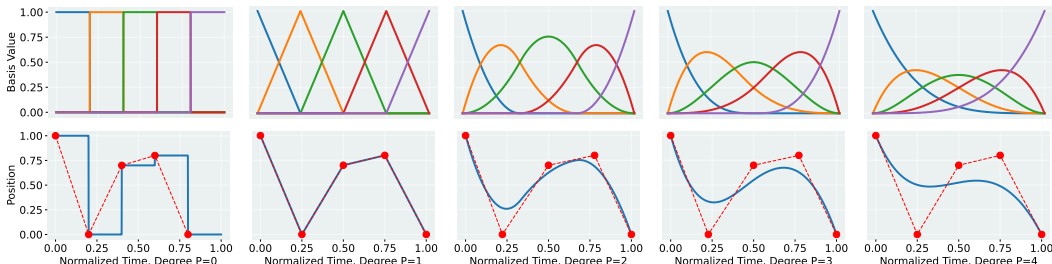

Figure 1: **From left to right**: Clamped B-Spline Basis $P = 0, 1, 2, 3, 4$ (top) and their generated trajectories (Bottom). Given the same **control points**, a higher degree will lead to smoother trajectories. All generated trajectories start exactly from the first control point and end at the last control point. Notably, action chunk is conceptually equivalent to B-Splines of 0-th degree, i.e., split-wise constants, as shown in the leftmost subplots. This relation is explained in details later in Section 4.1.

use k-means clustering to form discrete action bins, combined with residual offsets via separate prediction heads. VQ-BeT [15] extends this idea by encoding action chunks into codebook vectors using a Residual VQ-VAE [15]. While expressive, these methods require training encoder-decoder networks, which increases system complexity and introduces sensitivity to hyperparameters and quantization loss. In contrast, BEAST requires no additional tokenizer training and avoids such instabilities through direct B-spline representation. BEAST does not require any additional tokenizer training and does not increase training complexity through its direct B-spline representation.

More recently, FAST [18] proposes a compression-based tokenization strategy using discrete cosine transform and byte-pair encoding [42], resulting in fewer tokens per action chunk. As a consequence, the resulting action sequences can have varying lengths. This can complicate parallel decoding during inference. In comparison, BEAST produces fixed-length action representations. Fixed-length representations at every inference step allow for parallel decoding, significantly speeding up inference. OpenVLA-OFT [20] investigates how different tokenization strategies impact inference speed and policy performance in VLAs, showing that parallel decoding and action chunking can indeed lead to faster inference. However, OpenVLA-OFT does not compress the action tokens themselves, predicting an individual token for each action. BEAST compresses entire action chunks into a small set of B-spline control points. This enables both faster decoding and smooth, high-fidelity trajectories.

## 3 Preliminaries

**Problem Formulation.** Our goal is to train a policy $\pi(\boldsymbol{a}_{1:T} | \boldsymbol{s})$ that capable of mapping a given state $\boldsymbol{s}$ to a corresponding sequence of actions $\boldsymbol{a}_{1:T}$ which has $T$ time steps and $D$ Degrees of Freedom (DoF). To make this sequence prediction problem compatible with discrete generative models, we first transform the continuous action sequence into a sequence of discrete tokens. The goal of action sequence tokenization is to obtain a discrete token sequence $\bar{\boldsymbol{v}}_{1:J}$, where each token belongs to a vocabulary $\bar{\mathcal{V}}$ with size $|\bar{\mathcal{V}}|$, by defining a transformation $\texttt{tokenizer} : \boldsymbol{a}_{1:T} \rightarrow \bar{\boldsymbol{v}}_{1:J}$,

**B-Splines** (Basis Splines) [43] are widely used in the field of computer graphics and computer-aided design. A B-Spline curve $y$ is formulated through a linear basis function representation

$$\textbf{1-DoF B-Spline:} \qquad y(u) = \sum_{n=0}^{N-1} \Phi_n^P(u)\, c_n = \boldsymbol{\Phi}^P(u)\, \boldsymbol{c}, \quad 0 \le P < N, \quad u \in [k_0, k_M], \quad (1)$$

where $\boldsymbol{c}$ are $N$ *control points* and $u$ is a continuous parameter, often interpreted as *normalized time*. The *basis functions* $\boldsymbol{\Phi}^P(u) = [\Phi_0^P(u), .., \Phi_{N-1}^P(u)]$ are $N$ polynomial basis functions of $P$-th degree. These basis functions are defined over $M$ intervals determined by $M + 1$ knots in a *vector* $[k_0, ..., k_M]$, and it satisfies $M = N + P$ [43]. Typically, the knot vector is normalized such that $k_0 = 0$ and $k_M = 1$. The basis functions $\Phi_n^P(u)$ are recursively computed using the *Cox–de Boor recursion* [44]. We denote all *recursive degrees*[2] as $q = 0 : P$. For $q = 0$, the basis functions are

---

[2]The B-Spline degree $P$ differs from the recursive degree $q$. Trajectories are represented by basis functions of degree $P$, while lower recursive degree $q$ serve as intermediate representations in the recursive process.

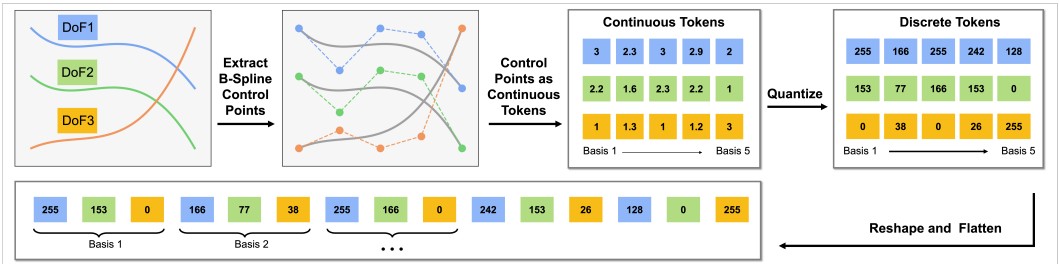

Figure 2: **Overview of the BEAST Encoding Pipeline:** Given a normalized action sequence, the BEAST pipeline first uses linear regression to extract continuous-valued control points, forming control point matrices that serve as intermediate continuous representations. These matrices are then quantized uniformly into discrete values within the range $[0, 255]$ and subsequently flattened to produce discrete action tokens for auto-regressive next-token prediction or parallel prediction.

defined as piecewise constant and recursively using the $(q-1)$-th degree basis for $q > 0$ with

$$\text{piecewise constant:} \quad \Phi_n^0(u) = \begin{cases} 1 & \text{if } k_n \leq u < k_{n+1}, \\ 0 & \text{otherwise.} \end{cases} \quad \text{and} \tag{2}$$

$$\text{recursive:} \quad \Phi_n^q(u) = k_n^{q-1}\Phi_n^{q-1}(u) + (1 - k_{n+1}^{q-1})\Phi_{n+1}^{q-1}(u), \tag{3}$$

where $k_n^{q-1} = (u - k_n)/(k_{n+q} - k_n)$.

**Clamped B-Spline.** In this work, we employ the *clamped uniform B-Spline*, where the first and last $P + 1$ knots are repeated to ensure that the resulting curve starts at the first control point and ends at the last control point. In Figure 1, we demonstrate the resulting basis functions of degrees from $P = 0$ to $P = 4$, together with their generated trajectories, given the same five control points. Clamped uniform B-splines are particularly suited for trajectory generation due to their smoothness, compact representation, and local support, where each control point only affects the curve locally.

**Parallel Decoding**. Unlike autoregressive generation, which predicts tokens sequentially and thus requires $K$ forward passes for a sequence of length $K$, *parallel decoding* [20] enables the prediction of the entire output sequence in a single forward pass. This is achieved by feeding the model with $K$ empty token embeddings and replacing the causal attention mask with a bidirectional mask, allowing the decoder to infer the entire sequence simultaneously. OpenVLA-OFT [20] leverages this approach for action sequence generation. In this work, we adopt the parallel decoding strategy to predict all BEAST tokens in a single pass, improving the inference efficiency without sacrificing accuracy.

## 4 B-Spline Encoded Action Sequence Tokenizer

In this section, we first describe how BEAST utilizes B-Spline to construct an efficient action sequence tokenizer that converts action sequences into either continuous or discrete action tokens. We then explain how smooth transitions between consecutive action sequences are achieved by enforcing the initial conditions of clamped B-splines. Finally, we discuss strategies for efficient integrating BEAST with various model architectures that predict discrete or continuous tokens.

### 4.1 Action Sequence Tokenization with B-Spline Tokenizer

Following prior works in action tokenization [12, 18], we first normalize the input actions such that the 1st and 99th quantile value of each action dimension in the dataset maps to the range of $[-1, 1]$. Using quantiles makes the normalization robust against outlier data points.

Figure 2 presents an overview of the tokenization process. We begin by considering the tokenization of a 1-DoF trajectory. Given a normalized action sequence $a_{1:T} = [a_1, a_2, ..., a_T]$ of length $T$, our goal is to determine a set of $N$ control points $\boldsymbol{c}$, with $N \leq T$, that approximate the given action sequence at spline evaluations $y(u)_{1:T}$. The linear transformation $u = t/T$ maps from action timestep to the parametric coordinate of the B-Spline. The spline evaluations $y(u)_{1:T}$ are approximated by minimizing the least-squares error

$$\boldsymbol{c} = \arg\min_{\boldsymbol{c}} ||y_{1:T} - a_{1:T}||_2^2 = \arg\min_{\boldsymbol{c}} ||\boldsymbol{\Phi}^P(u)\boldsymbol{c} - a_{1:T}||_2^2, \tag{4}$$

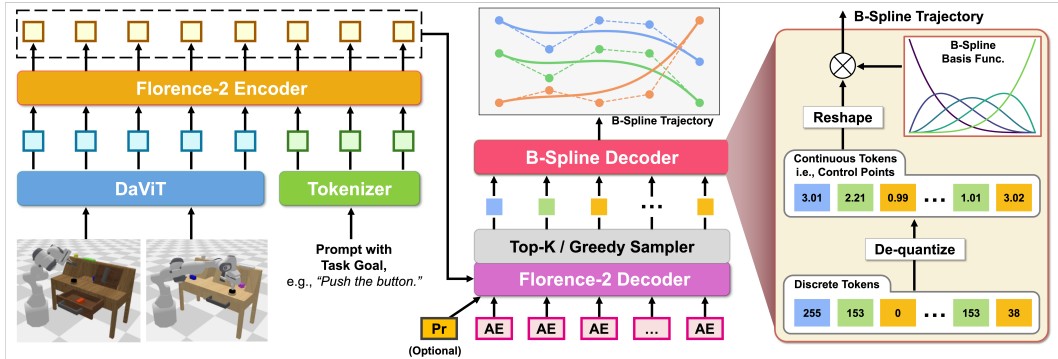

Figure 3: **BEAST-F** is a new VLA model that combines BEAST encoding with Florence-2 [45], a lightweight VLM with 0.77B parameters. BEAST produces uniform-length tokens, which allows BEAST-F to perform *parallel decoding* via learnable action embeddings (**AE**), instead of autoregressive next-token prediction. These discrete tokens are fed into the *B-Spline Decoder*, which first maps them to real-valued control points and then transforms those control points into continuous action sequences. The **Pr** token denotes an optional proprioceptive state.

where $\boldsymbol{\Phi}^P(u) = [\Phi_1^P(u), \Phi_2^P(u), ..., \Phi_N^P(u)]^\top$ represents precomputed B-spline basis functions defined over interval $u \in [0, 1]$. Ridge regression estimates the control points in closed form, $\boldsymbol{c} = [c_0, c_1, ..., c_{N-1}] = (\boldsymbol{\Phi}^\top \boldsymbol{\Phi} + \lambda \boldsymbol{I})^{-1} \boldsymbol{\Phi}^\top a_{1:T}$, with $\lambda$ acting as a regularization parameter. This efficient computation typically introduces only a small overhead, typically 3 to 5 milliseconds per batch. For a high-dimensional action sequence, i.e. $D > 1$, each DoF is encoded independently into $\boldsymbol{c}_d$, resulting in a matrix $\boldsymbol{C}$ of shape $D \times N$, that stacks each DoF's control points, $\boldsymbol{C} = [\boldsymbol{c}_1, \boldsymbol{c}_2, ..., \boldsymbol{c}_D]^\top$.

To form the final token sequence, this matrix is flattened by interleaving different action dimensions corresponding to the same basis functions, as illustrated in Figure 2. This flattening strategy preserves the temporal order inherent in the trajectory segments associated with each basis function.

**Remark 1: Connection to Action Chunking.** Action chunking, defined as a discrete sequence of actions $a_0, a_1, ..., a_T$, is mathematically equivalent to a piecewise constant function generated by 0-th degree B-splines. As demonstrated in Figure 1 left most, each action step $a_t$ can be identified as a control point $c_n$ of 0-th degree B-Spline basis with $t = n, T = N$.

## 4.2 Enforcing Smooth Transition with Clamped B-Spline

Executing long-horizon tasks typically requires producing multiple small action sequences that connect seamlessly (replanning). While predicting action sequences effectively improves consistency within individual action chunks, a significant challenge lies in managing discontinuities at transitions between consecutive chunks, which often result in jerky motion during online execution. Common approaches to address this issue apply temporal ensembles of actions [6, 46], calculating moving averages over multiple predictions. However, these temporal ensembles require high-frequency replanning (typically every timestep) to generate sufficient chunks for effective ensemble averaging, which significantly constrains execution speed in online applications.

In contrast, BEAST employs clamped B-Spline to ensure smooth transitions between consecutive action chunks. As introduced in Section 3, clamped B-Spline is a specialized variant of B-Spline that guarantees to start from the first control point and end at the last control point, which is utilized to generate seamlessly connected action sequences, as illustrated in Figure 1. To ensure smooth transitions, we directly set the first control point $\boldsymbol{c}_0$ to the last action of the previous sequence. We then compute the residual trajectory $\hat{\boldsymbol{a}}$ by subtracting the contribution of the first basis function: $\hat{\boldsymbol{a}} = \boldsymbol{a} - c_0 \Phi_0^P$. The remaining control points $\hat{\boldsymbol{c}} = [c_1, c_2, ...c_{N-1}]$ are determined by solving the linear regression problem similar to equation 4: $\arg\min_{\hat{\boldsymbol{c}}} ||\hat{\boldsymbol{\Phi}}^P(u)\hat{\boldsymbol{c}} - \hat{\boldsymbol{a}}||_2$. Through this approach, BEAST consistently generates action sequences with mathematically guaranteed smooth transitions between chunks. This will be further discussed in our toy task experiment in Section 5.1.

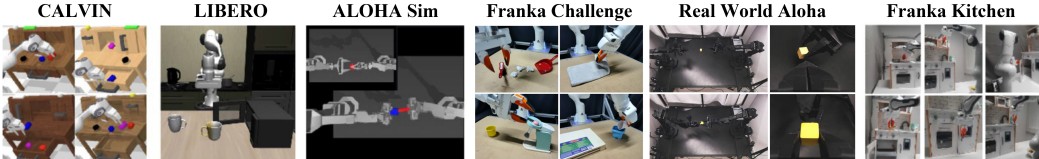

| CALVIN | LIBERO | ALOHA Sim | Franka Challenge | Real World Aloha | Franka Kitchen |

Figure 4: Simulation [6, 47, 48] and real world (Franka Challenge, Aloha, Franka Kitchen) tasks.

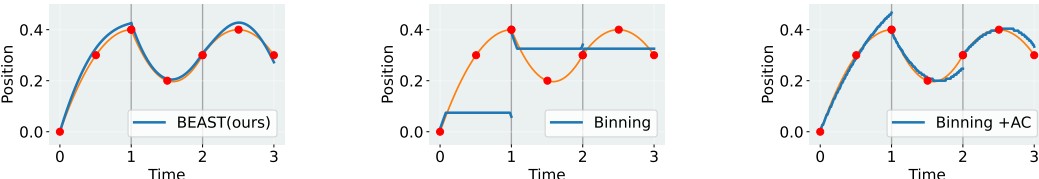

Figure 5: Comparison among BEAST, single-step binning tokenization and binning tokenization with action chunking (AC). The comparison is conducted through the same auto-regressive model with different tokenizers to fit the same **ground truth** cube splines given the same **context points**. BEAST is smooth within each sequence and continuous at the transitions between sequences.

### 4.3 Combining BEAST tokens with different architectures

**Discrete Tokens.** We first evaluate BEAST in a simplified setting with a decode-only transformer (see Figure 10) with CLIP [49] for language encoding and Film-conditioned ResNet-18 [50, 51] as image encoder. Film-ResNets are used in many prior works given their high efficiency and strong performance [52, 53, 13] The proprioceptive state of robot is projected to the embedding dimension with a two-layer MLP. We employ parallel decoding with bi-directional attention to accelerate the inference. To further demonstrate the scalability of BEAST with large-pretrained models, we combine BEAST with Florence-2, a small, pretrained VLM with Encoder-Decoder architecture (0.77B parameters). Following the previous works on autoregressive VLAs [12, 18], we overwrite the least used 256 tokens in the VLM vocabulary as our action tokens. We also employ a parallel decoding technique for the Florence variant, which significantly improves the throughput and reduces the latency for action generation. We provide an in-depth overview in Figure 3.

**Continuous Tokens.** We also explore the performance of combining BEAST with ACT [6]. ACT uses a conditional VAE (CVAE) with a Transformer Encoder-Decoder to predict a sequence of actions. We predict $N$ BEAST continuous tokens (normalized B-Spline control points without quantization), where each token has the dimension of $D$, instead of action sequences, this design choice keeps the temporal order inherent in the trajectory segments. Using BEAST, we reduce the length of predicted token sequence by 6.67 times (from 100 to 15) without sacrificing the task performance. In addition, our method enables smooth trajectories without requiring temporal aggregation.

## 5 Experiments

We conducted extensive evaluations in both simulated and real-world settings, targeting answering five key research questions (RQs): 1) What advantages does BEAST offer over commonly used binning-based tokenizers? 2) How does BEAST contribute to the performance on imitation learning benchmarks? 3) How does BEAST affect the training and inference efficiency? 4) Does BEAST generalize to real-world scenarios? 5) How do the design choices affect the performance of BEAST? BEAST is integrated into three different architectures: First, we combine BEAST and Florence-2 [45] and term this VLA variant as **BEAST-F**. Second, we integrate BEAST into a small decoder-only transformer trained from scratch, referred to as **BEAST-D**. Finally, we employ continuous BEAST tokens on top of a vanilla ACT[6], resulting in (**BEAST-ACT**). A detailed description of each architecture is provided in Appendix B. In contrast to many baselines, we test BEAST-F **without second-stage pretraining on large-scale robot datasets**.

| Train→Test | Method | PrT | Action Type | VLM | \multicolumn No. Instructions in a Row (1000 chains) | | | | | Avg. Len. |
|---|---|---|---|---|---|---|---|---|---|---|
| | | | | | 1 | 2 | 3 | 4 | 5 | |
| ABC→D | Diff-P-CNN [7] | × | Diffusion | × | 63.5% | 35.3% | 19.4% | 10.7% | 6.4% | 1.35 |
| | MDT [24] | × | Diffusion | × | 63.1% | 42.9% | 24.7% | 15.1% | 9.1% | 1.55 |
| | OpenVLA [12] | ✓ | Discrete | ✓ | 91.3% | 77.8% | 62.0% | 52.1% | 43.5% | 3.27 |
| | 3DDA [54] | × | Diffusion | × | 93.8% | 80.3% | 66.2% | 53.3% | 41.2% | 3.35 |
| | MoDE [53] | ✓ | Diffusion | × | 96.2% | 88.9% | 81.1% | 71.8% | 63.5% | 4.01 |
| | VPP [55] | ✓ | Diffusion | × | 95.7% | 91.2% | 86.3% | 81.0% | **75.0%** | 4.29 |
| | **BEAST-F (ours)** | × | Discrete | ✓ | **99.8%** | **96.5%** | **89.3%** | **82.7%** | 74.4% | **4.42** |
| ABCD→D | Diff-P-CNN [7] | × | Diffusion | × | 86.3% | 72.7% | 60.1% | 51.2% | 41.7% | 3.16 |
| | MoDE [53] | ✓ | Diffusion | × | 97.1% | 92.5% | 87.9% | 83.5% | 77.9% | 4.39 |
| | MDT [24] | × | Diffusion | × | **98.6%** | 95.8% | 91.6% | 86.2% | 80.1% | 4.52 |
| | **BEAST-F (ours)** | × | Discrete | ✓ | 98.1% | **96.2%** | **93.0%** | **89.3%** | **84.8%** | **4.61** |

Table 1: **CALVIN Benchmark results for ABC and ABCD.** The table reports average success rates for individual tasks within instruction chains and the average rollout length (Avg. Len.) to complete 5 consecutive instructions, based on 1000 chains. Zero standard deviation indicates methods without reported standard deviations. BEAST-F achieves SoTA performance in both tasks.

| | Spatial | | Object | | Goal | | Long | | Average | |
|---|---|---|---|---|---|---|---|---|---|---|
| | SR (↑) | Rank (↓) | SR (↑) | Rank (↓) | SR (↑) | Rank (↓) | SR (↑) | Rank (↓) | SR (↑) | Rank (↓) |
| Diff-P-CNN | 78.3 ± 1.1% | 6 | 92.5 ± 0.7% | 4 | 68.3 ± 1.2% | 6 | 50.5 ± 1.3% | 6 | 72.4 ± 0.7% | 6 |
| Octo | 78.9 ± 1.0% | 5 | 85.7 ± 0.9% | 6 | 84.6 ± 0.9% | 4 | 51.1 ± 1.3% | 5 | 75.1 ± 0.6% | 5 |
| OpenVLA | 84.7 ± 0.9% | 4 | 88.4 ± 0.8% | 5 | 79.2 ± 1.0% | 5 | 53.7 ± 1.3% | 4 | 76.5 ± 0.6% | 4 |
| $\pi_0$ | **96.8%** | 1 | **98.8%** | 1 | **95.8%** | 1 | 85.2% | 2 | 94.2% | 1 |
| $\pi_0$-FAST | 96.4% | 2 | 96.8% | 3 | 88.6% | 3 | 60.2% | 3 | 85.5% | 3 |
| **BEAST-F** | 92.9 % | 3 | 97.5 % | 2 | 93.1 % | 2 | **86.4 %** | 1 | 92.5% | 2 |

Table 2: **Experimental Results for the LIBERO Benchmarks.** SR: Success Rate. Best results in each column are shown in bold. BEAST-F achieves comparable performance state-of-the-art VLA, despite with a much smaller model and without robot data pretraining.

## 5.1 Comparing Against Binning-Based Tokenization

To answer **RQ1**, we begin with a 1D toy task to investigate the advantages of BEAST over binning-based tokenization. We follow the autoregressive prediction pipeline used in previous works [12, 18]. Note that BEAST can be used for both autoregressive prediction and parallel decoding. A small decoder-only transformer is trained to predict cubic splines from 3 control points. We compare against: 1) Single-step binning (denoted as **Binning**) [12], which discretizes each action into one of 256 bins, and 2) Chunk-level binning (denoted as **Binning+AC**), which discretizes entire action sequences of fixed length. We generate 2000 trajectories, 1s each at 100Hz resolution. Each model is trained for 8k steps and evaluated on 200 test sequences. BEAST achieves the lowest MSE ($0.0004 \pm 0.0005$),

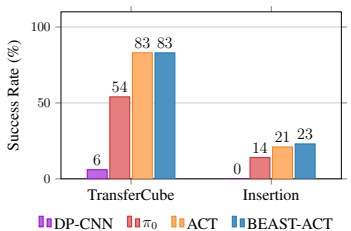

Figure 6: **ALOHA Benchmark results.** The success rate is reported over 500 episodes of evaluation.

outperforming chunked binning ($0.0009 \pm 0.0013$) and single-step binning ($0.0215 \pm 0.0216$), with the latter performing two orders of magnitude worse. To simulate real-world action chunking [7], we repeat the rollout prediction three times. As visualized in Figure 5, single-step binning fails to capture temporal structure and produces erratic outputs. Chunked binning captures some temporal coherence but results in jerky transitions due to discretization and a lack of continuity across chunks. In contrast, BEAST generates smooth trajectories with minimal error and requires only 5 tokens per 100-step sequence, resulting in an approximately 20x reduction in inference steps.

## 5.2 Strong Performance on Established Simulation Benchmarks

To answer **RQ2**, we evaluate BEAST on established simulation benchmarks and compare with other SoTA imitation learning methods and VLAs.

**Simulation Benchmarks.** CALVIN [47] features 34 tabletop manipulation tasks with a Franka Panda robot using delta end-effector control across four scene configurations (splits A-D). The dataset contains $24,000$ language-annotated demonstrations. We evaluate two settings: CALVIN ABC (zero-shot generalization) and CALVIN ABCD (scaling with more data). Performance is measured by success rates on sequential tasks and mean sequence length completion. All evaluations require policies to follow free-form language instructions and complete 5 tasks in sequence across $1,000$

different instruction chains. **LIBERO** [48] tests a delta-EEF controlled Panda Robot across various scenes with **130** diverse tasks. We report results on four specialized benchmark settings with 10 tasks each (Long, Spatial, Object, and Goal). Success is measured as the percentage of successful task completions across 50 trials per task. **ALOHA** [6] tests an absolute joint position controlled ALOHA Robot in two challenging bi-manual manipulation tasks that require high precision.

**Baselines.** We compare our Vision-Language-Action Model (VLA) against SOTA VLA policies and specialized approaches, using results reported in prior publications for fair comparison. Our primary baselines are OpenVLA [12] (7.7B parameters), $\pi_0$ [56] (3.3B parameters), $\pi_0$-FAST[18] (3.3B parameters), and a standard Diffusion Policy using a CNN [7]. For the bi-manual manipulation tasks, we compare the BEAST-ACT variant with small action chunking models to a vanilla ACT [6], $\pi_0$, and a standard Diffusion Policy using a CNN.

**Results.** Table 1 summarizes the performance of all policies on the CALVIN benchmark, where BEAST-F outperforms a diverse set of baselines across two settings, establishing a new state of the art. Unlike the most competitive baselines, BEAST-F achieves these results without relying on additional pretraining. On the various LIBERO benchmarks, our tokenizer achieves strong performance, being surpassed only by $\pi_0$-VLA. However, $\pi_0$ relies on large-scale pretraining to reach its performance, whereas BEAST-F remains competitive without it. In the most challenging long-horizon task setting, LIBERO-LONG, BEAST-F outperforms all baselines. See Table 2 for detailed results. For the bi-manual tasks (Figure 6), BEAST-ACT and ACT demonstrate significantly better performance than $\pi_0$. BEAST-ACT achieved a higher success rate than vanilla ACT in both tasks.

## 5.3 Advantages in Training and Inference Speed

Next, we verify the inference and training efficiency of BEAST to answer **RQ3**. Specifically, we consider the VLA variant **BEAST-F** and compare it against several recent VLAs [12, 56, 18], as well as a standard CNN-based Diffusion Policy[7]. We measure the inference efficiency on an RTX 4090 GPU. As shown in Table 3, BEAST-F demonstrates clear computational advantages. It achieves a throughput of

| Method | Throughputs (Hz)↑ | Latency (s)↓ |
|---|---|---|
| DP (0.26B) | 130.67 | 0.341 |
| OpenVLA (7B) | 6.09 | 0.164 |
| $\pi_0$ (3.3B) | 288.11 | 0.104 |
| **BEAST-F** (0.77B) | **617.3** | **0.019** |

Table 3: **Mean inference efficiency** (1000 steps in Bf16). All policies except OpenVLA use chunking length 50 (48 for DP).

617.3 Hz (e.g., generates approximately 617 actions per second), which is $2.14\times$ faster than $\pi_0$, $4.72\times$ faster than Diffusion Policy, and $101.4\times$ faster than OpenVLA. In addition, BEAST-F achieves the lowest latency at just 19 milliseconds, where latency refers to the time taken to generate one action chunk. These gains are due to the parallel decoding, which enables generating the action sequence in a single forward pass.

We further evaluate the training efficiency by comparing BEAST-F against $\pi_0$ and $\pi_0$-FAST. To exclude the bias introduced by the pretraining datasets, we trained all models **without robot dataset pretraining**. We report the success rate on LIBERO-LONG benchmark every 10k training steps in Figure 7. BEAST-F reaches a approximate $80\%$ success rate at just 20k steps, whereas $\pi_0$ reaches only around $20\%$ at the same point. Notably, $\pi_0$-FAST shows no success till 30k steps. $\pi_0$-FAST's poor performance indicates a heavy reliance on robot dataset pretraining, which further underscores the training efficiency of our method.

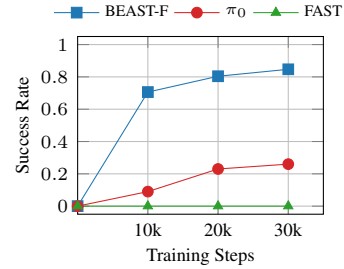

Figure 7: LIBERO-LONG.

## 5.4 Tokenizer Comparison under a Unified Backbone

To ensure a fair comparison between different tokenizers without the confounding effects of varying backbone architectures and different pretraining robot dataset pretraining recipes. We integrate FAST tokenization and binning tokenization (used by OpenVLA) into the Florence-2 model [45] without robot dataset pretraining. This setup enables an unbiased assessment of tokenizer performance. Both FAST and binning tokenizers are trained with autoregressive objective (FAST-AR and

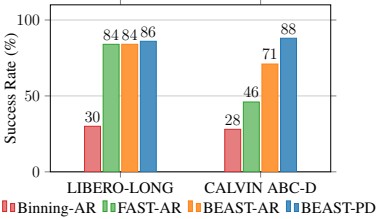

Figure 8: Tokenizer Comparison.

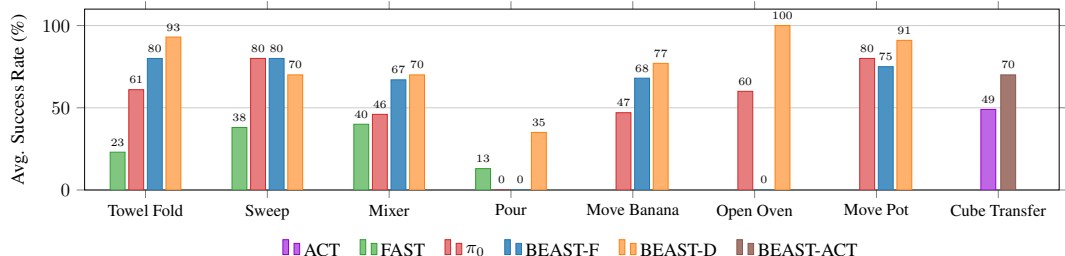

Figure 9: **Experimental Results on Real-World Robot Tasks.** This figure shows the average task success rate across eight real-world tasks. Each task and method was evaluated over 10 runs (30 runs for Cube Transfer). Success rates are measured at the sub-task level. Detailed descriptions of all sub-tasks are provided in Appendix F. BEAST variants achieve strong performance in real world.

Binning-AR), following their original implementations. Figure 8 presents results on the challenging LIBERO-Long benchmark and the Calvin ABC→D generalization benchmark. The BEAST tokenizer with parallel decoding (BEAST-PD) achieves the highest task completion ratio in both benchmarks, followed by BEAST with autoregressive decoding (BEAST-AR). BEAST-AR matches FAST on LIBERO-long and outperforms it on Calvin ABC→D. Overall, both compression-based tokenizers (BEAST and FAST) consistently outperform the binning tokenizer across all evaluations.

## 5.5   Real-World Evaluation with 3 Different Robot Setups

To answer **RQ4**, we assess the effectiveness of BEAST across diverse real-world scenarios with varying data collection frequencies. We evaluate BEAST on 8 challenging manipulation tasks across 3 different experimental setups: 1) **Franka Challenge**: Four tabletop manipulation tasks (Towel Fold, Sweep, Mixer, Pour) using a joint position-controlled Franka robot with data collected at 20Hz, 2) **Real Kitchen**: Three manipulation tasks on a toy kitchen setup (Move Banana, Open Oven, Move Pot) with data collected at 35Hz, 3) **Bi-manual ALOHA**: A cube transfer task using a bi-manual ALOHA robot with data recorded at 60Hz. For each task in the **Franka Challenge** and **Franka Kitchen** setups, we conduct 10 evaluation runs per method, while for the **ALOHA cube transfer** task, we performed 30 runs. The average success rate for each task is reported in Figure 9. For tasks comprising multiple stages, we track intermediate milestones to better evaluate the completion of each sub-task. Appendix F provides a detailed description of all setups and tasks. We compare BEAST against $\pi_0$[56], $\pi_0$-FAST[18], and ACT [6]. We finetune $\pi_0$ and $\pi_0$-FAST from the official pretrained checkpoints for an additional 60k and 40k steps, respectively. For each method, we train one multitask model for all four tasks, Real Franka tasks, and another for the Real Kitchen tasks. The results demonstrate that BEAST-F achieves $52.86\%$ success rate and BEAST-D achieves $76.57\%$. In contrast $\pi_0$ achieves $53.43\%$ and FAST only $28.5\%$. Interestingly, the smaller model (BEAST-D) outperforms all the VLAs, including the Florence variant with BEAST. We attribute this effect to the relatively small real-world dataset of only 50 demonstrations for each task. For the Aloha Cube Transfer task, we compare BEAST-ACT against the base ACT that directly predicts action sequences in the joint space. BEAST-ACT achieves $70\%$ success, which is $21\%$ higher than the base ACT.

## 5.6   Ablation Studies

To answer **RQ5**, we conduct ablation studies to analyze the impact of various design choices of BEAST. All experiments in this section use the Florence variant of BEAST and are evaluated on the CALVIN ABC benchmark. All results are summarized in Table 4.

**BEAST vs. Binning-based Tokenizer.** We first compare BEAST against a commonly used binning-based tokenizer in VLAs[12], which discretizes single-step actions into one of 256 uniformly distributed bins. We implement this baseline using the same Florence-2 backbone and denote it as **Binning-F**. It is trained to perform autoregressive token prediction. As shown in Table 4, BEAST significantly outperforms the binning-based approach, improving the average sequence length from $1.41$ to $4.43$, underscoring the effectiveness of BEAST as an action tokenizer.

| Variant | Avg. Len. |
|---|---|
| **BEAST-F (10)** | **4.43** |
| BEAST-F (5) | 3.88 |
| BEAST-F (15) | 4.20 |
| BEAST-F (20) | 4.32 |
| BEAST-F (25) | 4.23 |
| BEAST-SF | 3.98 |
| BEAST-CT | 3.93 |
| Binning-F | 1.41 |

Table 4: Average Sequence Lengths for BEAST-F Ablation.

**Discrete Tokens vs. Continuous Tokens.** Next, we study the choices between using discrete tokens or continuous tokens (denoted as **BEAST-CT**) as the action representation. In the continuous variant, the final hidden states of the Florence decoder are directly mapped to continuous BEAST tokens via a linear layer, and the learning objective is changed from cross-entropy to L1 regression loss. Results show that discrete tokens yield 12.7% better performance. We attribute this to the greater expressiveness of discrete representations, which are better suited to model multi-modal distributions.

**Choice of Number of Basis Functions.** Next, we evaluate how the number of basis functions affects the policy performance. We evaluate using $N = [5, 10, 15, 20, 25]$ basis functions to model action chunks of 20 steps, denoted as **BEAST-F [N]** in Table 4. Fewer basis functions lead to fewer tokens for prediction, but it also reduces the expressiveness of the B-Spline representation. On the contrary, more basis functions increase representational power but reduce compression, which can also negatively influence the performance.

**Scaling with Model Size.** Finally, we assess the impact of model size on task performance. We compare BEAST-F, which uses Florence-2-large (0.77B parameters), with **BEAST-SF**, a smaller variant based on Florence-2-base (0.23B parameters). The larger model achieves an 11.3% improvement in average sequence length, demonstrating that BEAST benefits from increased model capacity. This result highlights its potential as a scalable building block for larger VLAs.

# 6    Conclusion

We present BEAST, a B-spline–based tokenizer for continuous robot actions that compresses arbitrary trajectories into fixed-length token sequences while preserving smooth transitions between segments. BEAST supports discrete and continuous outputs and integrates seamlessly with various model architectures. By exploiting parallel decoding, it delivers fast inference and high compression rates without sacrificing performance. In extensive experiments—both in simulation and on real robots—BEAST consistently achieves strong results, demonstrating the effectiveness of our tokenization strategy.

**Limitations:** Although BEAST delivers strong performance, it is sensitive to the choice of the number of B-spline basis functions, which can markedly affect task outcomes (Section 5.6). The optimal count depends on the smoothness and sampling frequency of the trajectory; our experiments indicate that using 5–10 bases works well for one-second robot trajectories. A heuristic for selecting the number of basis functions is provided in Appendix A.

**Future Work**: We plan to extend BEAST to large-scale robot pretraining and to integrate continuous token representations with diffusion- and flow-matching objectives, aiming to further boost downstream task performance. Preliminary results on combining continuous BEAST tokens with a flow-matching objective are provided in Appendix D.

# 7    Acknowledgment

This work was supported by the pilot program Core Informatics of the Helmholtz Association (HGF), by the German Research Foundation (DFG, grant no. 448648559), and in part by the Helmholtz Association of German Research Centers. The authors also acknowledge support by the state of Baden-Württemberg through the HoreKa supercomputer funded by the Ministry of Science, Research and the Arts Baden-Württemberg, and by the German Federal Ministry of Education and Research.

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

# A    Choice of Number of Basis Functions

For finding the optimal N, we recommend starting with a number of basis functions equal to half of the sequence length. In practice, we typically sample a batch of action sequences from the dataset (e.g., 100 trajectories or more) and compute the reconstruction mean square error (MSE). An MSE below 1e-2 (for action sequences that are normalized into the range $[-1, 1]$) usually indicates that the chosen number of basis functions allows BEAST to represent the original trajectories well. We then repeat this process to find the minimal number of basis functions that achieves this threshold, thereby maximizing compression. Since BEAST does not require separate tokenizer training, this hyperparameter tuning process can be done in a few minutes across different action spaces and frequencies.

# B    Architectures

**BEAST-F** is a new VLA model that integrates BEAST with the pretrained VLM Florence-2 [45]. Florence-2 is a compact vision-language model with 0.77B parameters, featuring an encoder-decoder language model architecture paired with a DaViT image encoder[57] that efficiently compresses images into sequences into just 50 tokens. Florence-2 offers two unique advantages: First, Florence-2 was pretrained with a large-scale dataset focusing on image-grounding with tasks like bounding box prediction and image segmentation. These objectives are well aligned with robotic manipulation challenges. In addition, its small size and low image token count make it computational and memory efficient and enable us to run VLA experiments on consumer-grade hardware. This combination of task-relevant pretraining and computational efficiency made Florence-2 ideal for exploring our tokenizers within practical resource constraints.

**BEAST-D** is a compact Transformer model that integrates discrete BEAST tokens. The architecture is shown in Figure 10.

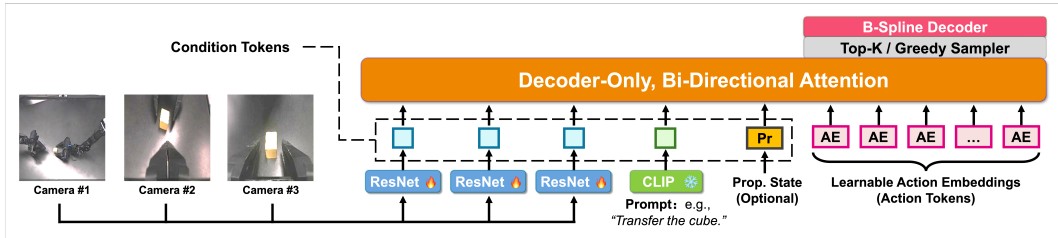

Figure 10: **Overview of BEAST-D**: BEAST-D is a small transformer model that integrates BEAST. It replaces the causal attention in the decoder-only transformer with bidirectional attention to enable fast parallel decoding. BEAST-D integrates ResNet as image encoder and CLIP as language encoder.

# C    Baselines Implementation

$\pi_0$: $\pi_0$ is a generalist robot policy that combines a pre-trained VLM backbone with a lightweight *action expert* module trained from scratch to generate continuous actions using *flow matching*. A key innovation of $\pi_0$ is its cross-embodiment training strategy, which integrates over 900M timesteps of data from 7 distinct robot embodiments and 68 manipulation tasks, enabling generalization across heterogeneous hardware platforms. The model is trained using a two-phase pipeline: a large-scale pre-training stage leveraging Internet-scale semantic priors, followed by post-training on curated task-specific data to enhance performance on complex, dexterous tasks.

**FAST**: FAST introduces a novel compression-based tokenization method, named *Frequency-space Action Sequence Tokenization*, for training autoregressive VLA models on high-frequency, dexterous robot control tasks. Unlike prior VLAs that struggle with discretizing continuous actions at high frequencies, FAST leverages the *Discrete Cosine Transform (DCT)* and *Byte-Pair Encoding (BPE)* to produce compact, information-rich action tokens, marking a significant advance in training efficiency.

# D Preliminary Results with Flow Matching

We conduct an additional experiment on combining BEAST's continuous tokens with flow-matching loss[58, 59] and a mixture-of-expert backbone[53]. The resulting model, BEAST-Flow achieves performance comparable to the Flow policy with action chunking on LIBERO-Long, and outperforms it on Calvin-ABC, while predicting only half as many tokens. The full results are presented in Figure 11. These results highlight the effectiveness of BEAST within a flow-matching framework.

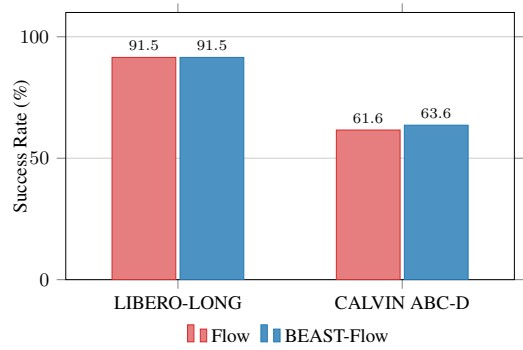

Figure 11: Preliminary Results with Flow Matching.

# E Hyperparameters

| Hyperparameter | LIBERO | | | | CALVIN | |
| --- | --- | --- | --- | --- | --- | --- |
| | SPATIAL | OBJECT | GOAL | LONG | ABCD→D | ABC→D |
| Action Sequence Length | 20 | 20 | 20 | 20 | 20 | 20 |
| Number of Basis | 10 | 10 | 10 | 10 | 10 | 10 |
| Vocabulary Size | 256 | 256 | 256 | 256 | 256 | 256 |
| Optimizer | AdamW | AdamW | AdamW | AdamW | AdamW | AdamW |
| Betas | [0.9, 0.95] | [0.9, 0.95] | [0.9, 0.95] | [0.9, 0.95] | [0.9, 0.95] | [0.9, 0.95] |
| Learning Rate | 2e-5 | 2e-5 | 2e-5 | 2e-5 | 2e-5 | 2e-5 |
| Batch Size | 128 | 128 | 128 | 128 | 32 | 32 |
| Train Steps (k) | 35 | 35 | 50 | 70 | 30 | 30 |

Table 5: Summary of BEAST-F hyperparameters for all simulation experiments.

| Hyperparameter | REAL KITCHEN | REAL FRANKA |
| --- | --- | --- |
| Action Sequence Length | 80 | 20 |
| Number of Basis | 15 | 5 |
| Vocabulary Size | 256 | 256 |
| Optimizer | AdamW | AdamW |
| Betas | [0.9, 0.95] | [0.9, 0.95] |
| Learning Rate | 2e-5 | 2e-5 |
| Batch Size | 96 | 96 |
| Train Steps (k) | 60 | 60 |

Table 6: BEAST-F hyperparameters for real robot experiments.

| Hyperparameter | REAL KITCHEN | REAL FRANKA |
|---|---|---|
| Action Sequence Length | 80 | 20 |
| Number of Basis | 10 | 5 |
| Vocabulary Size | 256 | 256 |
| Transformer Layers | 6 | 6 |
| Attention Heads | 8 | 8 |
| Embedding Dim | 256 | 256 |
| Image Encoder | FiLM-ResNet18 | FiLM-ResNet18 |
| Goal Lang Encoder | CLIP ViT-B/32 | CLIP ViT-B/32 |
| Attn Dropout | 0.1 | 0.1 |
| Residual Dropout | 0.1 | 0.1 |
| MLP Dropout | 0.1 | 0.1 |
| Optimizer | AdamW | AdamW |
| Betas | [0.9, 0.999] | [0.9, 0.999] |
| Learning Rate | 3e-4 | 3e-4 |
| Weight Decay (Trans/Other) | 0.05 / 0.05 | 0.05 / 0.05 |
| Batch Size | 384 | 256 |
| Train Steps (k) | 60 | 60 |
| EMA | False | False |

Table 7: BEAST-D hyperparameters for real robot experiments.

# F    Real Robots Setup & Tasks

## F.1    Robot System Details

**Real Kitchen**. This setup consists of a single Franka Emika robot operating within a simulated kitchen environment. It is equipped with two OAK-D Lite cameras providing top-down and side perspectives, each delivering visual input at a resolution of 250×250 pixels. The robot has an 8-dimensional configuration and action space, which includes seven joint and one gripper states.

**Real Franka**. This configuration features a single Franka Emika robot situated in a general-purpose tabletop environment designed for more challenging manipulation tasks. Visual observations are obtained from two Orbbec Femto Bolt cameras, positioned to capture left and right perspectives. The input images are resized to a resolution of 180×320 pixels. The robot configuration and action space remain the same as the Franka Kitchen setup.

**ALOHA**. Based on the ALOHA setup [6], the system incorporates two 6-DoF Trossen ViperX robotic arms. The environment includes two wrist-mounted and an additional top-mounted Logitech C920 camera. The combined system operates in a 14-dimensional configuration and action space, accounting for both arms' joint and gripper states.

## F.2    Tasks Description and Evaluation Metrics

In the Real Kitchen setup, the robot performs pick-and-place tasks, whereas in the Real Franka setup, the robot is required to execute more diverse manipulation behaviors, such as sweeping or pouring. For each task performed by the Franka Emika robot, a scoring rubric is defined to quantitatively evaluate task progression. The specific evaluation criteria for each task are detailed below.

- **Open the door:** The task begins under one of two initial conditions: with or without an object placed on the stove. The objective is for the robot to open the oven door. Task completion is evaluated as a success or a failure. Although the task involves three key motion phases, as shown in Figure 12 (Open the door), all the policies under evaluation are capable of completing the task in its entirety once the robot successfully grasps the handle. A trial is considered successful if the robot fully opens the oven door by first grasping the handle and then using its fingers to push the opposite side of the door, ensuring that it is completely open. We conducted four evaluation trials with no object on the stove and one with a randomly placed object.

- **Banana into the pot:** In this task, the robot aims to grasp a banana and place it on or into a pot. The initial conditions are categorized based on the relative positions of the pot and the

banana, as well as the position of the banana relative to the corresponding stove. The pot is assumed to be correctly positioned on the stove. The banana, however, may be placed directly on the stove in one of three orientations or slightly offset to the left or right. A total of 10 trials are conducted across these different initial configurations. Task performance is scored on a scale of 0 to 3, with one point awarded for each of the following criteria: (1) successfully positioning the banana between the robot's two fingers, (2) lifting the banana off the surface, and (3) placing the banana onto or into the pot. If the robot attempts to grasp more than three times, exhibits jerky hand movements, or significantly displaces the pot from its original position, the subtask is awarded 0.5 points to reflect partial completion.

- **Pot into the sink:** This task is similar to the one described previously. The initial conditions in this task differ based on the relative position between the pot and the two stoves. Task performance is evaluated using three criteria: (1) successfully positioning the pot between the robot's two fingers - note that directly grasping the pot with its handle is considered an unstable grasp and is awarded 0.5 points, (2) lifting the pot off the surface, and (3) placing the pot in the sink. In this task, penalties for jerky hand movements are still applied.

- **Towel folding:** The objective of this task is to neatly fold a towel that is randomly oriented at the beginning of each trial. One point is awarded for each of the following: lifting a corner of the towel, completing the fold, and achieving accurate alignment of the folded towel.

- **Sweep:** In the Sweep task, the positions of the broom, dustpan, and trash vary across trials. Four pieces of trash are placed on the table for the robot to clean. A maximum of four points can be awarded, based on the following criteria: successfully grasping the broom, performing a single sweeping motion, demonstrating the ability to execute multiple sweeping motions, and sweeping all trash into the dustpan.

- **Mixer:** In this task, a cup and a mixer are placed on the table. The robot's objective is to sequentially (1) open the mixer, (2) grasp the cup, (3) place the cup on the mixer's platform, and (4) close the mixer. Task performance is evaluated based on the successful completion of these four subtasks, with one point awarded for each. Notably, unlike previous tasks, the language instructions provided to the robot consist of three separate sentences corresponding to the actions of opening/closing the mixer and placing the cup onto the platform.

- **Pour:** In the Pour task, the source cup is initially placed on a platform and contains plastic pellets that simulate liquid. The objective is to pour the pellets into a designated target cup. Task performance is evaluated out of a maximum of 4 points, awarded based on the following criteria: (1) successfully grasping the source cup, (2) pouring the pellets into the target cup, (3) ensuring that all pellets are poured into the target cup, and (4) placing the source cup back on the platform.

- **ALOHA cube transfer:** In the cube transfer task, the ALOHA robot is designed to pick up a randomly placed cube using its right arm and then transfer the cube to its left arm. The performance of the task is evaluated by assigning scores to three specific steps: (1) successfully picking up the cube, (2) successfully initiating the transfer with the left arm making contact with the cube, and (3) the left arm successfully taking possession of the cube while the right arm releases it.

## G   Compute Resources

We train and evaluate all the models based on our private clusters. Each node contains 4 NVIDIA A100, for BEAST-F we use 4 GPUs for training. For BEAST-D and BEAST-ACT, we use one GPU for training. We report the average training cost in Table 8.

|  | BEAST-F | BEAST-D | BEAST-ACT |
|---|---|---|---|
| vRAM | 64GB | 8GB | 15GB |
| steps/hour | 6000 | 10000 | 11000 |

Table 8: Training time for each variant.

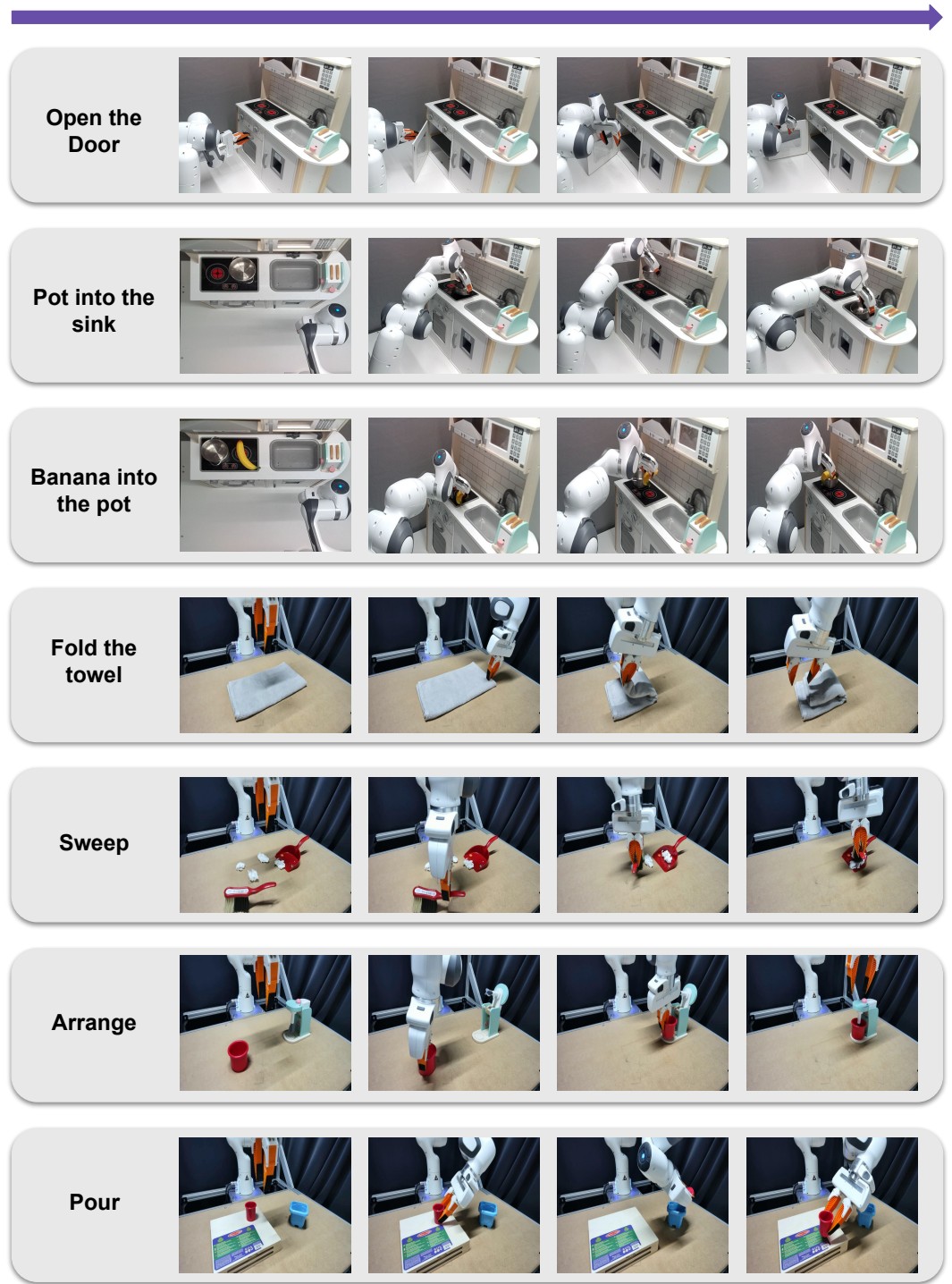

Figure 12: **Key frames for real world different tasks**

