# OpenReview forum: "BEAST: Efficient Tokenization of B-Splines Encoded Action Sequences for Imitation Learning"
_NeurIPS.cc/2025/Conference — NeurIPS 2025 poster_

### Official Review · Reviewer_s1wN · 2025-07-02

**Clarity:** 3
**Significance:** 3
**Originality:** 2
**Rating:** 4
**Confidence:** 1

**Summary:**

The paper introduces BEAST (B-spline Encoded Action Sequence Tokenizer), a novel method for representing continuous robot action sequences as compact, fixed-length discrete or continuous tokens. A core aspect of BEAST is its use of B-splines, which intrinsically ensures smooth, continuous trajectories without requiring separate tokenizer training.

The key contributions of this work are:
*   **Novel B-spline-based Tokenization:** BEAST encodes continuous action sequences using B-splines, providing a compact and efficient representation.
*   **Versatility Across Architectures:** The paper demonstrates BEAST's compatibility by integrating it with various model architectures, including VAEs (BEAST-ACT), decoder-only Transformers (BEAST-D), and Vision-Language Models like Florence-2 (BEAST-F), supporting both continuous and discrete token objectives.
*   **Enhanced Efficiency:** BEAST significantly reduces computational costs during both training and inference. Its fixed-length token sequences enable faster action generation through parallel decoding, requiring significantly fewer tokens compared to binning-based methods.
*   **Guaranteed Smoothness:** By leveraging clamped B-splines and enforcing initial conditions, BEAST ensures mathematically guaranteed smooth transitions between consecutive action chunks, preventing jerky movements.
*   **Strong Performance:** Extensive evaluations on 166 simulated tasks (CALVIN, LIBERO, ALOHA) and 8 real-world robotic tasks across three different setups demonstrate that BEAST achieves competitive or superior task success rates compared to state-of-the-art methods, often without relying on large-scale pretraining.
*   **Ablation Studies:** The paper includes thorough ablation studies analyzing the impact of design choices such as discrete versus continuous tokens, the number of basis functions, and model size, providing insights into BEAST's performance characteristics.

A noted limitation is BEAST's sensitivity to the chosen number of B-spline basis functions, which can impact performance.

**Questions:**

Here are some questions and suggestions for the authors, focusing on actionable points for a productive rebuttal:

1.  **Robustness and Generalization of Basis Function Selection (Line 337-339):**
    *   **Question:** The limitations section acknowledges sensitivity to the number of B-spline basis functions (`N`), stating that "The optimal count depends on the smoothness and sampling frequency of the trajectory; our experiments indicate that using 5-10 bases works well for one-second robot trajectories." However, Table 4 shows a significant drop in average sequence length (performance) from `N=15` (4.14) to `N=20` (2.71) for a 20-step action chunk (Line 322). This suggests that `N=20` might be suboptimal or that higher `N` values *can* negatively influence performance, which contradicts the intuition that more basis functions provide more representational power. Could the authors elaborate on the trade-offs beyond "compressing vs. representational power"? Is there a specific threshold or relationship between `N` and `T` (action sequence length) beyond which performance degrades significantly, or a methodology for finding the optimal `N` given varying trajectory characteristics?
    *   **Suggestion:** Expand on the reasons for the performance drop at `N=20` in Table 4. Discuss more deeply the practical considerations and a more systematic approach (e.g., a small search, heuristic, or adaptive method) for selecting `N` across different tasks and action sequence lengths, especially given the sensitivity mentioned.
    *   **Criteria for Score Change:** Providing a more detailed analysis of the `N` selection, perhaps with a clear methodology for practitioners to determine `N` or a deeper insight into the observed performance drop, would strengthen the paper's claims about its practical applicability. If the authors can show a more robust way to handle this sensitivity, the score would increase.

2.  **Comparison with Other Trajectory Parameterizations / Implicit Representations:**
    *   **Question:** The paper primarily compares BEAST against "binning-based tokenization" (Section 5.1) and other VLA/Diffusion policies that might use simpler action chunking or direct prediction. While BEAST's B-spline approach offers advantages like smoothness and fixed-length tokens, there are other established methods for continuous trajectory parameterization in robotics (e.g., Gaussian Process Motion Priors, Dynamic Movement Primitives, or even learned implicit representations for trajectories). Could the authors discuss how BEAST compares conceptually and/or empirically (if relevant prior work exists) against these alternative trajectory representations, especially concerning their ability to ensure smoothness, achieve compression, and integrate with deep learning architectures?
    *   **Suggestion:** Add a brief discussion in the Related Work or Limitations section acknowledging other continuous trajectory representations. Even a qualitative discussion of their pros and cons relative to BEAST's B-spline approach would add significant value by contextualizing BEAST within a broader spectrum of trajectory encoding methods.
    *   **Criteria for Score Change:** Acknowledging and briefly discussing other continuous trajectory parameterization methods would demonstrate a more comprehensive understanding of the landscape and position BEAST more clearly within it, leading to a higher score.

3.  **Implications of "Without Pretraining" (Line 247-248):**
    *   **Question:** The paper frequently highlights that BEAST-F achieves strong results "without relying on additional pretraining" (Line 247) or "without robot data pretraining" (Line 248). While this is a significant advantage, Florence-2 itself is a large pre-trained Vision-Language Model. Could the authors clarify the distinction more explicitly? Is the claim specifically that BEAST-F *doesn't require additional robot-specific pretraining* (beyond what Florence-2 already has for general vision-language tasks), or that it performs well *from scratch* (which seems less likely given Florence-2's nature)? A nuanced explanation would prevent potential misinterpretations regarding the extent of "pretraining" involved.
    *   **Suggestion:** Rephrase or elaborate on the "without pretraining" claim to clearly state that it refers to *robot-specific* or *second-stage* pretraining on large robot datasets, distinguishing it from the foundational pretraining of Florence-2. This would make the contribution clearer and more precise.
    *   **Criteria for Score Change:** A precise clarification of what "without pretraining" entails in the context of BEAST-F, acknowledging the base VLM's pretraining, would increase the clarity and accuracy of the paper's claims, leading to a higher score.

**Ethical Concerns:**

["NO or VERY MINOR ethics concerns only"]

**Final Justification:**

Thank you for the detailed rebuttal, which has addressed most of my concerns. I will raise the scores.

**Limitations:**

Please check the weakness section.

**Paper Formatting Concerns:**

Good format

**Quality:**

2

**Strengths And Weaknesses:**

### Strengths:

*   **Originality:** The core idea of using B-splines for action sequence tokenization, specifically the direct encoding without separate training and the inherent smoothness guarantee, appears novel in the context of imitation learning and VLA models. This differentiates it from common binning, VQ-VAE, or compression-based methods.
*   **Quality - Technical Soundness:** The mathematical formulation of B-splines and their application to action sequences (Equation 4, clamped B-splines for smooth transitions) is well-grounded. The claim of mathematically guaranteed smoothness is a strong technical advantage. The efficiency gains from fixed-length tokens and parallel decoding are clearly articulated and supported by empirical results.
*   **Quality - Empirical Performance:** The paper presents extensive evaluations across numerous simulated benchmarks (CALVIN, LIBERO, ALOHA, 166 tasks) and real-world robot setups (8 tasks across 3 different robots). Consistently achieving competitive or state-of-the-art results, often with smaller models and without large-scale robot-specific pretraining, demonstrates the method's effectiveness. The ablation studies provide useful insights into design choices.
*   **Clarity:** The paper is generally well-written and easy to follow. The introduction clearly outlines the problem and BEAST's proposed solution. Figure 1 and 2 effectively illustrate the core concepts of B-splines and the encoding pipeline. The experimental setup and results are presented clearly with appropriate tables and figures.
*   **Significance:** BEAST offers a practical solution to several challenges in robot imitation learning, including action sequence discretization, ensuring smooth trajectories, and improving inference efficiency. Its compatibility with various architectures (VAE, Transformer, VLM) makes it a versatile tool. The efficiency gains (2.14x to 101.4x faster inference, faster training convergence) are highly significant for real-world robotic applications and research. The reduced token count is also a notable contribution to efficiency.

### Weaknesses:

*   **Quality - Sensitivity to Basis Function Selection (Actionability):** While the paper acknowledges the sensitivity to the number of B-spline basis functions (\`N\`) in the limitations, the explanation for why \`N=20\` performs significantly worse than \`N=15\` for a 20-step chunk in Table 4 is somewhat vague ("reduce compression, which can also negatively influence the performance"). This is a critical hyperparameter, and the guidance "5-10 bases works well for one-second robot trajectories" might not generalize well to different action chunk lengths or robot dynamics. A more principled or robust way to select \`N\`, or a deeper understanding of its impact beyond a simple trade-off, is missing.
*   **Clarity - Notation for B-spline Parameters:** There's a slight potential for confusion between the B-spline degree \`P\` and the number of control points \`N\` (e.g., "Remark 1" stating \`T=N\` for \`P=0\` vs. \`N <= T\` for \`P\`-th degree, and how \`N\` is precisely chosen for \`P > 0\` degrees given a \`T\`-length action sequence). While explained, a concise clarification on the typical choices of \`P\` and \`N\` in relation to the action sequence length \`T\` in practice would improve understanding.

---

> ### Author Rebuttal · Authors · 2025-07-30
>
> We would like to thank the reviewer for taking the time to review our work and the many helpful comments and suggestions. We hope the following replies sufficiently address the raised questions and concerns. We will update the paper accordingly.
>
> > **Q1, W1:** …However, Table 4 shows a significant drop in average sequence length (performance) from `N=15` (4.14) to `N=20` (2.71) for a 20-step action chunk (Line 322). This suggests that `N=20` might be suboptimal or that higher `N` values *can* negatively influence performance, which contradicts the intuition that more basis functions provide more representational power. Could the authors elaborate on the trade-offs beyond "compressing vs. representational power"? Is there a specific threshold or relationship between `N` and `T` (action sequence length) beyond which performance degrades significantly, or a methodology for finding the optimal `N` given varying trajectory characteristics?
> >
>
> Thank you for pointing this out, that’s indeed a valuable observation. We investigated the irregular performance drop from `N = 15` to `N = 20` in more detail. We found that it was due to overfitting in the ridge estimator (line 145~146) when too many basis functions are used. Specifically, the default regularization parameter was set to a very small value ($\lambda=1e-9$), which we found insufficient.  After increasing it to $\lambda=1e-4$, BEAST-F with `N = 20` also achieves 4.32 on CALVIN ABC. To further validate this, we re-evaluated the the model for `N = 15`, and added an additional ablation for `N = 25`. The complete ablation is attached below.
>
> | Number of Basis | 5 | **10** | 15 | 20 | 25 |
> | --- | --- | --- | --- | --- | --- |
> | Calvin ABC | 3.88 | **4.43** | 4.20 | 4.32 | 4.23 |
>
> For finding the optimal `N`, we recommend starting with a number of basis functions equal to half of the sequence length. In practice, we typically sample a batch of action sequences from the dataset (e.g., 100 trajectories or more) and compute the reconstruction mean square error (MSE). An MSE below 1e-2 usually indicates that the chosen number of basis functions allows BEAST to represent the original trajectories well. We then repeat this process to find the minimal number of basis functions that achieves this threshold, thereby maximizing compression. Since BEAST does not require separate tokenizer training, this hyperparameter tuning process can be done in a few minutes across different action spaces and frequencies. We will include this practical tuning guide in the appendix.
>
> > **Q2:** The paper primarily compares BEAST against "binning-based tokenization" (Section 5.1) and other VLA/Diffusion policies that might use simpler action chunking or direct prediction. While BEAST's B-spline approach offers advantages like smoothness and fixed-length tokens, there are other established methods for continuous trajectory parameterization in robotics (e.g., Gaussian Process Motion Priors, Dynamic Movement Primitives, or even learned implicit representations for trajectories). Could the authors discuss how BEAST compares conceptually and/or empirically (if relevant prior work exists) against these alternative trajectory representations, especially concerning their ability to ensure smoothness, achieve compression, and integrate with deep learning architectures?
> >
>
> We appreciate the reviewer’s constructive suggestion. In response, we will include the following discussion in the related works section:
>
> Dynamic Movement Primitives (DMPs)[1] employ a second-order dynamic system with a goal attractor. When we use DMPs to encode action chunks, only trajectory position and velocity at the transitions are continuous, because of its second-order formulation. Additionally, their goal is only converged when time goes to infinity instead of finite time and its resulting position basis functions (after integrating the dynamic system) is asymmetric, as shown Fig. 1 in [2] , which potentially influences its representation capacity. In contrast, the B-Splines representation allows for arbitrary-order smoothness by increasing the spline degree and the basis functions are symmetric, as reported in Fig. 1 top row, in our paper.
>
> To tackle the issues of DMPs, recent works [3, 4] have also explored using B-splines as a Movement Primitives representation. However, these methods mainly focus on reinforcement learning settings and only utilize it as continuous action abstraction.
>
> **To our best knowledge, BEAST is the first work that integrates B-splines into imitation learning architectures, generating discrete tokens for VLAs.**
>
> Gaussian Process Motion Planner (GPMPs) [5] identifies waypoints as support states and uses Gaussian Process (GP) to interpolates the trajectories.  However, GP-based methods are usually constrained to conditioned on low-dimensional conditions, such as via-points. It is unclear how to employ them with high-dimensional conditions, such as images and language. To overcome this issue, Conditional Neural Movement Primitives (CNMPs) [7] was introduced to model trajectories using Conditional Neural Processes (CNPs) [6]. CNMPs need a separate training process and cannot guarantee continuous transitions (see Fig 10 (b) in [2]). **In contrast, BEAST requires no separate training for tokenization and guarantee smooth transitions between chunks.**
>
> We summarize a comparison of these works in the table below:
>
> | Model | Need training | Chunk transition smoothness | High dimensional conditioning (Language, image) |
> | --- | --- | --- | --- |
> | DMP [1, 2] | No | Position, Velocity | Yes [2] |
> | GPMP [5] | No | No | No |
> | CNMP (CNP) [6, 7] | Yes | No | Yes [7] |
> | BEAST (ours) | No | Arbitrary orders | Yes |
>
> > **Q3:** The paper frequently highlights that BEAST-F achieves strong results "without relying on additional pretraining" (Line 247) or "without robot data pretraining" (Line 248). While this is a significant advantage, Florence-2 itself is a large pre-trained Vision-Language Model. Could the authors clarify the distinction more explicitly? Is the claim specifically that BEAST-F *doesn't require additional robot-specific pretraining* (beyond what Florence-2 already has for general vision-language tasks), or that it performs well *from scratch* (which seems less likely given Florence-2's nature)? A nuanced explanation would prevent potential misinterpretations regarding the extent of "pretraining" involved.
> >
>
> We thank the reviewer for pointing this out, for the “without relying on additional pretraining” we indeed mean without additional robot-specific pretraining. We would highlight this even more in the final version.
>
> > **W2: Clarity - Notation for B-spline Parameters:** There's a slight potential for confusion between the B-spline degree `P` and the number of control points `N` (e.g., "Remark 1" stating `T=N` for `P=0` vs. `N <= T` for `P`-th degree, and how `N` is precisely chosen for `P > 0` degrees given a `T`-length action sequence). While explained, a concise clarification on the typical choices of `P` and `N` in relation to the action sequence length `T` in practice would improve understanding.
> >
>
> We thank the reviewer for pointing out the confusion in the notation. We will update the paper accordingly.
>
> In the BEAST tokenizer, there are three key hyperparameters:
>
> - `T`: the length of an action chunk (i.e., number of actions),
> - `P`: the degree of the B-spline, which determines the level of continuity of the trajectory,
> - `N`: the number of basis functions, i.e., control points per action dimension.
>
> Here we attach a table to demonstrate the relation between P and level of continuity:
>
> | P (Degree) | 0 | 1 | 2 | 3 | 4 |
> | --- | --- | --- | --- | --- | --- |
> | Continuity | None | Position | Pos+Vel | Pos+Vel+Acc | Pos+Vel+Acc+Jerk |
>
> In BEAST, we typically choose `P = 4` to achieve continuity up to jerk trajectory.
>
> The number of basis functions `N` determines how many control points are used for each action dimension within one action chunk. A higher `N` offers higher granularity in trajectory representation but risks overfitting to noise in action sequences. Smaller `N`, on the other hand, offers better compression but risks underfitting and dexterity loss.
>
> According to the recursive definition of B-splines, `N` must satisfy `N > P` [8].
>
> Mathematically, there is no hard dependencies between T and N. In practice, using `N ≤ T` can compress token sequence length, resulting in more efficient training and inference. For details on how the optimal value of `N` is determined, please refer to our response to Q1.
>
> To your question regarding Remark 1, when choosing `P = 0` and `N = T`, B-splines are equivalent to action chunking, where each control point correspond exact to an action point.
>
> ### References:
>
> **[1]** Ijspeert A. et al., "Dynamical movement primitives: learning attractor models for motor behaviors," *Neural Computation* 2013.
>
> **[2]** Li G. et al., "Prodmp: A unified perspective on dynamic and probabilistic movement primitives," *IEEE RA-L* 2023.
>
> **[3]** Liao W. et al., "BMP: Bridging the Gap between B-Spline and Movement Primitives," *2nd CoRL Workshop on Learning Effective Abstractions for Planning*, 2024.
>
> **[4]** Kicki P. et al., "Bridging the gap between Learning-to-plan, Motion Primitives and Safe Reinforcement Learning," *CoRL* 2024.
>
> **[5]** Mukadam M. et al., "Gaussian process motion planning," *IEEE ICRA* 2016.
>
> **[6]** Garnelo M. et al., "Conditional neural processes," *ICML* 2018.
>
> **[7]** Seker M. Y. et al., "Conditional Neural Movement Primitives," *RSS* 2019.
>
> **[8]** Prautzsch H. et al., *Bézier and B-spline Techniques*, Springer, Aug. 2002.

---

> ### Author Response · Authors · 2025-08-05
>
> As the discussion period is ending soon, we would greatly appreciate it if you could let us know whether our response addressed all of your remaining concerns. If you have any remaining questions or points that require clarification, we are happy to provide additional information. Thanks a lot!

---

### Official Review · Reviewer_JQnF · 2025-07-14

**Clarity:** 3
**Significance:** 4
**Originality:** 3
**Rating:** 5
**Confidence:** 4

**Summary:**

The paper introduces BEAST (B-spline Encoded Action Sequence Tokenizer), a novel method for tokenizing continuous action sequences in imitation learning. Instead of relying on existing tokenization methods like vector quantization or binning, BEAST employs B-spline basis functions to compress continuous action trajectories into fixed-length, smooth representations. These can be used as discrete tokens (after quantization) or continuous control points, enabling compatibility with both autoregressive and parallel decoding architectures.

Key contributions include:

1. A tokenizer that does not require separate training and produces fixed-length, smooth action sequences.
2. Compatibility with multiple model types (e.g., transformer, CVAE, Florence-2).
3. Strong empirical performance across simulation benchmarks (CALVIN, LIBERO) and real-world robotic tasks.
4. Substantial improvements in training/inference efficiency via parallel decoding and action compression.

**Questions:**

1. Section 5.5 shows that BEAST is sensitive to the number of basis functions. Can you provide heuristics or general rules for selecting this parameter across different robot platforms and frequencies?

2. Are there scenarios where BEAST fails to outperform traditional tokenizers, such as in highly discontinuous or impulsive control settings?

3. You highlight reduced token counts. Can you elaborate on any observed tradeoffs between compression and policy expressivity beyond MSE?

**Ethical Concerns:**

["NO or VERY MINOR ethics concerns only"]

**Limitations:**

Yes.

**Paper Formatting Concerns:**

In line 320 and Table 4., the indices are incorrectly processed as hyperlinks to references.

**Quality:**

4

**Strengths And Weaknesses:**

# Quality

**Strengths**: novel and practical method that improves both performance and efficiency; extensive experimental validation, including simulation and real-world benchmarks; strong ablations showing the value of B-spline representation and design choices; impressive improvements in inference speed (2-100x over baselines); smooth trajectory generation is demonstrably better than binning or quantized chunking.

**Weaknesses**: some ablation results (e.g., choice of basis functions) suggest non-trivial sensitivity, but there's limited discussion on tuning strategies; paper could better detail failure cases or conditions where BEAST might underperform

# Clarity

**Strengths**: the methodology is well-explained, especially the B-spline encoding and its advantages; figures (e.g., tokenization pipeline, spline smoothness) are informative; experimental setup and benchmarks are clearly described.

**Weaknesses**: the distinction between discrete and continuous token modes in some sections could be more explicit; descriptions of integration into large models like Florence-2 may assume familiarity with these architectures.


# Significance

**Strengths**: addresses a core bottleneck in modern VLAs-tokenization of continuous actions; broadly applicable to robotics, VLMs, and imitation learning; effective across multiple benchmarks and robots without large-scale pretraining; could lower the barrier to applying VLAs in real-world robotics.

# Originality

**Strengths**: B-spline-based tokenization is novel in this domain; clever exploitation of B-spline properties (local support, smooth transitions, fixed-length encoding); avoids training a separate tokenizer: conceptually elegant and practically useful.

**Weaknesses**: some concepts (e.g., B-spline smoothing, ridge regression) are not new, though their application here is novel.

---

> ### Author Rebuttal · Authors · 2025-07-30
>
> We would like to thank the reviewer for the positive evaluation of our work and the many helpful comments and suggestions. We hope the following replies sufficiently address the raised questions and concerns. We will update the paper accordingly.
>
> > W1: some ablation results (e.g., choice of basis functions) suggest non-trivial sensitivity, but there's limited discussion on tuning strategies; paper could better detail failure cases or conditions where BEAST might underperform.
> >
>
> Thank you for pointing this out, that’s indeed a valuable observation. We investigated the irregular performance drop from `N = 15` to `N = 20` in more detail. We found that it was due to overfitting in the ridge estimator (line 145~146) when too many basis functions are used. Specifically, the default regularization parameter was set to a very small value ($\lambda=1e-9$), which we found insufficient.  After increasing it to $\lambda=1e-4$, BEAST-F with `N = 20` also achieves 4.32 on CALVIN ABC. To further validate this, we re-evaluated the the model for `N = 15`, and added an additional ablation for `N = 25`. The complete ablation is attached below.
>
> | Number of Basis | 5 | **10** | 15 | 20 | 25 |
> | --- | --- | --- | --- | --- | --- |
> | Calvin ABC | 3.88 | **4.43** | 4.20 | 4.32 | 4.23 |
>
> > W2.1: the distinction between discrete and continuous token modes in some sections could be more explicit;
> >
>
> We apologize for the lack of clarity. To clarify: **continuous tokens** refer to the normalized B-spline control points used directly, without any quantization. In contrast, **discrete tokens** are obtained by discrete these control points into uniform bins ranging from 0 to 255. We will update the paper accordingly.
>
> > W2.2: Descriptions of integration into large models like Florence-2 may assume familiarity with these architectures.
> >
>
> We agree with the reviewer that a short description of Florence-2 will enhance the clarity, therefore we will include the following description into the appendix for the final version:
>
> Florence-2 [1] is a compact vision-language model with 0.77B parameters, featuring an encoder-decoder language model architecture paired with a DaViT image encoder that efficiently compresses images into sequences into just 50 tokens. Florence-2 offers two unique advantages: First, Florence-2 was pretrained with a large-scale dataset focusing on image-grounding with tasks like bounding box prediction and image segmentation. These objectives are well aligned with robotic manipulation challenges. In addition, its small size and low image token count make it computational and memory efficient and enable us to run VLA experiments on consumer-grade hardware. This combination of task-relevant pretraining and computational efficiency made Florence-2 ideal for exploring our tokenizers within practical resource constraints.
>
> > Q1: Section 5.5 shows that BEAST is sensitive to the number of basis functions. Can you provide heuristics or general rules for selecting this parameter across different robot platforms and frequencies?
> >
>
> As we mentioned in our response to W1, the sensitivity was caused by an extremely small regularization parameter in the ridge regression, which has now been addressed. We agree with the reviewer that choosing an appropriate number of basis functions is crucial for the performance of BEAST. As a general heuristic, we recommend starting with a number of basis functions equal to half of the sequence length. In practice, we typically sample a batch of action sequences from the dataset (e.g., 100 trajectories or more) and compute the reconstruction mean square error (MSE). An MSE below 1e-2 usually indicates that the chosen number of basis functions allows BEAST to represent the original trajectories well. We then repeat this process to find the minimal number of basis functions that achieves this threshold, thereby maximizing compression. Since BEAST does not require separate tokenizer training, this hyperparameter tuning process can be done in a few minutes across different action spaces and frequencies.
>
> > Q2: Are there scenarios where BEAST fails to outperform traditional tokenizers, such as in highly discontinuous or impulsive control settings?
> >
>
> We assume the reviewer refers to traditional tokenizers such as single-step binning or quantized chunking (i.e., Binning + AC). Action chunking can be interpreted as a special case of zero-degree B-splines, and we provide further discussion on this in Section 4.1. In highly discontinuous or impulsive control scenarios, where sharp changes or sudden actions dominate, we can configure BEAST to replicate traditional tokenizer’s behavior. Specifically, setting the spline degree `P = 0` and number of basis functions `N = 1` allows BEAST to act as a single-step binning tokenizer, while setting the `P = 0` and  `N = T` emulates quantized action chunking. We include a table below for clarity:
>
> |  | **Spline Degree P** | **Number of Basis Functions N** |
> | --- | --- | --- |
> | Binning | 0 | 1 |
> | Binning + AC | 0 | T (Action Sequence Length) |
>
> Therefore, even in highly discontinuous or impulsive control settings, BEAST does not inherently fail to outperform traditional methods, instead, it includes them as special cases within its formulation. This highlights one of BEAST’s key strengths: its representational flexibility to adapt to a wide range of control characteristics.
>
> > Q3: You highlight reduced token counts. Can you elaborate on any observed tradeoffs between compression and policy expressivity beyond MSE?
> >
>
> We agree with the reviewer that there is a trade-off between compression and policy performance. Fewer basis functions yield smoother trajectories and shorter token sequences, improving compression. However, too few may underfit and miss essential trajectory features, resulting in sub-optimal policy performance, especially in tasks that require precise and dexterous control. This can be observed from the table in response to W1, where `N = 5` only achieves average sequence length of 3.88, which is noticeable lower than `N = {10, 15, 20, 25}`. On the other hand, more basis functions improve expressiveness but risks overfitting to noise present in the original action sequences and reduce compression gains. Notably, BEAST does not aim to solely minimize the MSE directly but prioritizes generating smooth trajectories with low MSE.
>
> > In line 320 and Table 4., the indices are incorrectly processed as hyperlinks to references.
> >
>
> Thanks for pointing out the typo, we will update the paper accordingly
>
> ## References:
>
> [1] Xiao, Bin, et al. "Florence-2: Advancing a unified representation for a variety of vision tasks." *CVPR* 2024.

---

> > ### Comment · Reviewer_JQnF · 2025-08-07
> >
> > Thanks for the authors' detailed rebuttal that has addressed all my concerns. I will maintain my positive ratings (5: Accept). Good work!

---

> > > ### Author Response · Authors · 2025-08-07
> > >
> > > We sincerely thank the reviewer for their positive feedback and constructive suggestions, which helped us improve the paper further. We are also pleased that our response addressed their concerns.

---

### Official Review · Reviewer_Ca1e · 2025-07-22

**Clarity:** 3
**Significance:** 3
**Originality:** 3
**Rating:** 4
**Confidence:** 4

**Summary:**

The paper proposes an action sequence modeling approach that is better suited for representing continuous actions and is more efficient in capturing temporal dynamics.

**Questions:**

When I read the introduction, I expect your work to address a concrete problem in VLA—specifically, the limitations of existing methods that use autoregressive models to predict discrete actions. However, when I get to Section 4.3 and the experiments, I no longer understand what problem your method is trying to solve. At one point, it is combined with ACT (which already predicts continuous actions rather than discrete ones), and at another point, it is compared with methods like Pi0, which use DIT to output continuous actions—even though the architectures are fundamentally different. I think you are trying to show your work is goot at lots of aspects, but such experiments design is weak to prove that.

That said, I think your idea is valuable. What I expect to see is whether your method can be plugged into existing frameworks—for example, replacing the action modeling strategy in Pi0 or OpenVLA—and bring consistent improvements. Instead, the current evaluation focuses on combining your method with DaViT and Florence to form Beast-VLA, but these vision-language models are rarely used in prior VLA work, which makes the comparison less convincing.

The following are detailed questions that may overlap with the overall points mentioned above:

- Why do you choose DaViT and Florence to build your VLA model, instead of using more commonly adopted backbones such as DINO or SigLIP?

- Why do you combine with ACT or replace ACT’s action modeling, given that ACT is unrelated to discrete action modeling?

- Why not integrate your method into PI0 or OpenVLA to demonstrate that your action modeling approach can improve existing VLA models? While PI0 uses DiT to predict continuous actions and is not based on discrete action modeling, it still serves as a strong and relevant baseline to show your action modeling strategy can even outperform DiT's action modeling.

**Ethical Concerns:**

["NO or VERY MINOR ethics concerns only"]

**Final Justification:**

After rebuttal, I keep my score as borderline accept

**Limitations:**

Yes

**Quality:**

3

**Strengths And Weaknesses:**

- Strengths：
    - The idea is both interesting and intuitive. It adopts a more explainable approach to action modeling and demonstrates how it improves the smoothness of transitions between action chunks. The method proves to be effective and efficient in the context of robotic manipulation.
-  Weaknesses:
   - The main weakness lies in the experimental design used to demonstrate how the proposed method works and its advantages. I believe this is partly due to the experimental setup and partly due to the writing. Please refer to the questions for more details.

---

> ### Author Rebuttal · Authors · 2025-07-30
>
> We would like to thank the reviewer for the positive evaluation of our work and the many helpful comments and suggestions. We hope the following replies sufficiently address the raised questions and concerns. We will update the paper accordingly.
>
> > Q1: Why do you choose DaViT and Florence to build your VLA model, instead of using more commonly adopted backbones such as DINO or SigLIP?
> >
>
> We chose DaViT as vision encoder because Florence-2 was pre-trained with DaViT, we maintained DaViT to preserve the learned multimodal alignment between visual and language representations. Replacing DaViT with another vision encoder like DINO or SigLIP could break this carefully learned alignment.
>
> We chose Florence-2 as our VLM backbone for two main reasons:
>
> 1. Florence-2 was pre-trained on a large-scale dataset focusing on image-grounding tasks, such as bounding box prediction and image segmentation. These objectives are well aligned with robotic manipulation tasks.
> 2. With only 0.77B trainable parameters, Florence-2 is significantly more compact and computationally  efficient than larger alternatives like PaliGemma (3B) used by pi0 [3] or Llama-2 (7B) by OpenVLA [5]. This makes it a practical backbone for our work, which emphasizes efficiency and tokenizer design rather than maximizing backbone performance.
>
> > Q2: Why do you combine with ACT or replace ACT’s action modeling, given that ACT is unrelated to discrete action modeling?
> >
>
> The idea is to demonstrate that BEAST can also be used to generate continuous tokens. While ACT already generates continuous actions, it typically requires temporal ensemble [1,2] across predictions to smooth trajectories. In contrast, the BEAST formulation inherently ensures trajectory smoothness and continuity in transitions. Through empirical experiments, we demonstrate that the BEAST-enhanced ACT achieves these benefits without compromising performance.
>
> > Q3: Why not integrate your method into PI0 or OpenVLA to demonstrate that your action modeling approach can improve existing VLA models? While PI0 uses DiT to predict continuous actions and is not based on discrete action modeling, it still serves as a strong and relevant baseline to show your action modeling strategy can even outperform DiT's action modeling.
> >
>
> We fully agree with the reviewer that integrating BEAST with PI0 and OpenVLA backbone would strengthen our argument. However, extremely high computational resources demand and closed-source, proprietary datasets make this kind of comparison extremely difficult. OpenVLA, for instance, employs Llama-2 7B as its backbone and was pretrained using **21,500 A100 GPU hours (equivalent to 14 days on a clusters of 64 A100s)** [5].  PI0 and PI0-FAST are pretrained on large closed-source,  proprietary dataset (they use a large pretrained dataset **~10000 hours with only 9.1% open-source data and 903M timesteps closed-source data** [3]). Furthermore,  the pretraining pipelines for PI0 and PI0-FAST are not publicly available, making integrating BEAST extremely challenge.
>
> To still provide a meaningful and fair comparison, we instead combined FAST tokenization[4] and OpenVLA tokenization[5] with Florence-2 model. This enables a fair assessment of performance between different tokenizers. Here are the results on the most challenging LIBERO-Long benchmark and Calvin ABC→D generalization benchmark. BEAST tokenizer with parallel decoding (BEAST-F-PD) achieves the highest task completion in both benchmark, followed by BEAST with autoregressive (BEAST-F-AR), yielding equal or better performance than FAST and OpenVLA tokenizers.
>
> |  | **BEAST-F-PD** | BEAST-F-AR | FAST-F | OpenVLA-F |
> | --- | --- | --- | --- | --- |
> | LIBERO-Long | **0.864** | 0.837 | 0.837 | 0.304 |
> | Calvin-ABC | **4.42** | 3.57 | 2.3 | 1.41 |
>
> Additionally, to simulate a minimal “PI0-style” alternative without requiring large-scale pretraining, we conduct an additional experiment on combining BEAST’s continuous tokens with flow-matching loss and a mixture-of-expert backbone [6]. The resulting model, **BEAST-Flow** achieves performance comparable to the Flow policy with action chunking on LIBERO-Long, and outperforms it on Calvin-ABC, while **predicting** **only half as many tokens**. The full results are presented in the table below. These results highlights the effectiveness of BEAST within a flow-matching framework.
>
> |  | **BEAST-Flow** | Flow |
> | --- | --- | --- |
> | LIBERO-Long | **0.915** | **0.915** |
> | Calvin-ABC | **3.18** | 3.08 |
>
> ### References:
>
> **[1]** Zhao T. et al., "Learning fine-grained bimanual manipulation with low-cost hardware," *RSS* 2023.
>
> **[2]** Li Q. et al., "Cogact: A foundational vision-language-action model for synergizing cognition and action in robotic manipulation," *arXiv preprint arXiv:2411.19650*, Nov. 2024.
>
> **[3]** Black K. et al., "$\pi_0$: A Vision-Language-Action Flow Model for General Robot Control,"  *RSS* 2025.
>
> **[4]** Pertsch K. et al., "FAST: Efficient action tokenization for vision-language-action models,"  *RSS* 2025.
>
> **[5]** Kim M. et al., "OpenVLA: An open-source vision-language-action model," CoRL 2024.
>
> **[6]** Reuss M. et al., "Efficient diffusion transformer policies with mixture of expert denoisers for multitask learning," *ICLR* 2025.

---

> > ### Comment · Reviewer_Ca1e · 2025-08-04
> >
> > Actually, before reading your response, I had already anticipated that the lack of comparison with Pi0 or OpenVLA was likely due to resource constraints. I understand this, but I believe it’s important to clearly state this limitation in the paper so that readers are aware of the reason. I will maintain my current score and am inclined to recommend acceptance.

---

> > > ### Author Response · Authors · 2025-08-04
> > >
> > > We thank the reviewer for the response and positive evaluation of our work. We will incorporate this discussion into the final version of the paper. We are happy to address any further questions or concerns that may arise during the discussion period.

---

### Author Response · Authors · 2025-08-09
**Thank you for the insightful discussions and help to improve BEAST!**

#

Dear Reviewers,

We sincerely thank all of the reviewers for the thoughtful feedback
and engaging discussions throughout the rebuttal process. Your insights
have been invaluable in helping us improve our work on BEAST. Going into
the reviewer discussion, we would like to summarize the main
improvements that resulted from your feedback during the rebuttal:

- **Related Work**: Expanded discussion on BEAST's relation to other continuous trajectory parameterization methods.
- **Limitation**: Elaborated on the reason for not evaluating BEAST with Pi0 and OpenVLA backbone.
- **Tokenizer Comparisons**: Added evaluations comparing BEAST, FAST, and OpenVLA (Binning) tokenizers with the same Florence-2 backbone,  showing BEAST’s strong performance against existing tokenizers.
- **BEAST + FLOW**: Included evaluations combining BEAST with the flow matching objective, demonstrating consistent gains in efficiency and policy performance.
- **Ablation on the Number of Basis Functions:** Addressed parameter sensitivity when increasing the number of basis functions and re-running the ablation with updated regularization parameters.
- **Heuristic for Basis Selection:**  Introduced a heuristic for choosing the number of basis functions to balance compression and expressiveness.
- **Writing Clarity:**  Improved explanations of continuous vs. discrete tokens, model architectures, and the definition of “additional pretraining.”, etc.

These changes have significantly strengthened our paper and all discussed points will be included in the final version.

Thank you again for your time and valuable feedback and we hope you take these points into account for your final decision.

Sincerely,

The Authors

---

### Decision · Program_Chairs · 2025-09-17

**Decision:**

Accept (poster)

**Comment:**

The paper proposes an interesting idea for tokenizing continuous action sequences in imitation learning, which compresses continuous action trajectories into fixed-length, smooth representations. It addresses a core bottleneck in modern VLAs, and proves effective across multiple benchmarks and robots without large-scale pretraining.

All reviewers vote for acceptance, and the AC accordingly recommends accepting this submission.